# Glucosylsphingosine (lyso-Gb1) as a Biomarker for Monitoring Treated and Untreated Children with Gaucher Disease

**DOI:** 10.3390/ijms20123033

**Published:** 2019-06-21

**Authors:** Noa Hurvitz, Tama Dinur, Michal Becker-Cohen, Claudia Cozma, Marina Hovakimyan, Sebastian Oppermann, Laura Demuth, Arndt Rolfs, Aya Abramov, Ari Zimran, Shoshana Revel-Vilk

**Affiliations:** 1Gaucher Unit, Shaare Zedek Medical Center, Hebrew University, Jerusalem 9103102, Israel; noa.hurvitz@mail.huji.ac.il (N.H.); dinurtama@gmail.com (T.D.); michalbec@gmail.com (M.B.-C.); abrahamova@szmc.org.il (A.A.); azimran@gmail.com (A.Z.); 2Faculty of Medicine, The Hebrew University of Jerusalem, Ein Kerem, Jerusalem 9112102, Israel; 3Centogene AG, Rostock 18055, Germany; claudia.cozma@centogene.com (C.C.); marina.hovakimyan@centogene.com (M.H.); sebastian.oppermann@centogene.com (S.O.); laura.demuth@centogene.com (L.D.), arndt.rolfs@centogene.com (A.R.); 4Faculty of Medicine, University of Rostock, Rostock 18051, Germany

**Keywords:** Gaucher disease, glucosylsphingosine, lyso-Gb1, biomarker, children

## Abstract

The role of glucosylsphingosine (lyso-Gb1), a downstream metabolic product of glucosylceramide, for monitoring treated and untreated children with Gaucher disease (GD) has not yet been studied. We reviewed the clinical charts of 81 children (<18 years), 35 with mild type 1 GD (GD1), 34 with severe GD1 and 12 with type 3 GD (GD3), followed at Shaare Zedek Medical Center between 2014–2018. Disease severity for GD1 was based on genotypes. Forty children (87%) with severe GD1 and GD3 received enzyme replacement therapy (ERT) compared to two children (6%) with mild GD1. Lyso-Gb1 measurements were conducted on dried blood spot samples taken at each clinic visit. Lyso-Gb1 levels were significantly lower in children with mild compared to severe GD1 (*p* = 0.009). In untreated children, lyso-Gb1 levels were inversely correlated with platelet counts. During follow-up, lyso-Gb1 increased in almost 50% of untreated children, more commonly in younger children. In treated children, lyso-Gb1 levels were inversely correlated with hemoglobin levels. The increase of lyso-Gb1 while receiving ERT, seen in eight children, was partly associated with compliance and weight gain. Lyso-Gb1 seems to be a useful biomarker for monitoring children with GD and should be included in the routine follow-up. Progressive increase in lyso-Gb1 levels in untreated children suggests ERT initiation.

## 1. Introduction

Gaucher Disease (GD) is among the most prevalent, recessively inherited, lysosomal storage disorder caused by a deficiency in the enzyme β-glucocerebrosidase. The deficient enzymatic activity, results in the lysosomal accumulation of its substrate glucosylceramide, most prominently in macrophages [1]. Glycosphingolipid-laden macrophages, referred to as Gaucher cells, accumulate in the visceral tissues liver, spleen and bone marrow, inducing a pleiotropic array of symptoms, including hepatosplenomegaly and pancytopenia, in addition to bone complications such as non-specific bone pain, bone crises, avascular necrosis and pathologic fractures [2]. Three variants of GD are generally distinguished based on the absence (GD 1) or the presence of central nervous system involvement (GD 2 or GD 3).

In the past, the majority of GD cases manifested during childhood and adolescence, with almost half of the cases diagnosed before ten years of age [3]. In early symptomatic disease, early intervention with enzyme replacement therapy (ERT) has become the standard of care since 1991 [4], and it has led to improved GD-related symptoms and significant reduction of complications (particularly bone crises and osteonecrosis) [5]. With the extended use of carrier screening for GD, the phenotype of children with GD had changed, and many of them are asymptomatic or mildly symptomatic at diagnosis [6,7]. Under these circumstances, the question arises to whom ERT should be given and when to initiate it, as some of those children will remain asymptomatic for many years, while others may develop irreversible complications that may be avoided by early initiation of ERT.

Glucosylsphingosine (lyso-Gb1), a deacylated form of glucosylceramide, is also degraded by the glucocerebrosidase. Lyso-Gb1 was proved to be a highly sensitive and specific biomarker for diagnosis and monitoring of adults patients with GD [8,9]. This is the first study reporting the role of lyso-Gb1 for monitoring disease status in a relatively large cohort of treated and untreated children with GD.

## 2. Results

The clinical phenotype of all children at their last visit is presented in Table 1. As expected, the majority of children with severe GD1 and GD3 received ERT, whereas the majority of children with mild GD1 were untreated. At the time of the last visit, lyso-Gb1 levels were significantly lower in children with mild GD1 compared to those with severe GD1 (*p* = 0.009) and not different from children with GD3 (*p* = 0.81). No significant differences were found between children with mild and severe GD1 regarding the age, gender, platelet count, hemoglobin levels and spleen and liver MN (multiples of normal) volume. Significantly larger spleen and liver MN volume were found in children with GD3 compared to those with GD1 (*p* = 0.007 and *p* = 0.005, respectively).

In all groups combined, the lyso-Gb1 significantly correlated with platelet count (*p* < 0.0001, *r* = −0.42) and hemoglobin levels (*p* = 0.003, *r* = −0.35), but not with liver MN volume, spleen MN volume, child’s age, and weight. When analyzing according to genotype, a significant correlation between lyso-Gb1 and platelet was found only in children with mild GD1 and a significant correlation between lyso-Gb1 and hemoglobin was found only in children with severe GD1. Similarly, when analyzing according to treatment status, irrespective of genotype, a significant inverse correlation was found between lyso-Gb1 and platelet count only in untreated children (*p* = 0.002 vs. *p* = 0.178 [in treated children]) and between lyso-Gb1 and hemoglobin level only in treated children (*p* = 0.01 vs. *p* = 0.197 [in untreated children]) (Figure 1).

Pre-ERT data, available for ten children, were compared to the first visit data of the 28 untreated children (Table 2). As expected, children eventually treated had more symptomatic disease, i.e., thrombocytopenia, anemia, and hepatosplenomegaly. Importantly, although lyso-Gb1 was not used for treatment decisions, significantly higher levels of lyso-Gb1 levels were found in the pre-ERT group compared to untreated children (*p* = 0.0003).

The patterns of change in lyso-Gb1 levels from baseline to the last visit for untreated children, treated children with pre-ERT measurements and treated children with both measurements on-ERT are shown in Table 3. In treated children, the drop of lyso-Gb1 was significantly higher in the pre-ERT group compared to the on-ERT group (*p* = 0.007) (Table 3). Lyso-Gb1 levels increased with time in eight children, all from the on-ERT group (Table 4). Retrospective analysis of the clinical charts found compliance issues and weight gain (>15%) without dose adjustment, as a possible explanation. Age was not associated with the pattern of change in treated children (Figure 2A).

In untreated children, lyso-Gb1 levels increased with time in almost half of the cohort (Table 3). The group of children whose lyso-Gb1 levels increased during follow-up were significantly younger at baseline compared to the group of children whose lyso-Gb1 levels dropped or remained unchanged during follow-up (*p* = 0.01) (Figure 2B).

Time of follow-up and weight change were not associated with the pattern of change of lyso-Gb1 levels in treated and untreated children.

## 3. Discussion

In this study, we show the potential value of lyso-Gb1 in children with GD both for monitoring untreated and treated children. An association between GD severity and lyso-Gb1 levels was expressed in several ways. First, an association between genotype and the lyso-Gb1 level was shown; significantly higher levels of lyso-Gb1 in children with severe GD1 compared to children with mild GD1 (Table 1). Second, we found an association between lyso-Gb1 levels and clinical status, i.e., platelet count, in untreated children (Figure 1A). Third, we show that children who eventually started treatment, based on clinical criteria, had significantly higher levels of lyso-Gb1 compared to untreated children (Table 2).

Association between genotype and lyso-Gb1 levels are consistent with adult data showing that patients with less severe disease, i.e., homozygotes N370S, generally show more modest increases in plasma lyso-Gb1 [10]. As expected, clinical variability was found within the group of N370S homozygous patients [11]. Within this group of N370S homozygous patients, a clear relationship between disease severity and lyso-Gb1 levels was observed by us and others [12]. With the change in the phenotype of children with GD, a useful biomarker to guide therapy decisions in untreated children is important.

In treated children, the lyso-Gb1 increase in some of the cases was associated with compliance and significant weight gain (>15%). We believe that the increase in lyso-Gb1 was related to lower ERT dose/kg and not the weight change per se. The linkage between the dose of ERT and response was previously shown in patients with GD [13].

Recently, we have shown in adults with GD1 that lyso-Gb1 is a reliable response biomarker preceding changes in other disease parameters [14]. Initiation of ERT or a substrate reduction agent had a significant effect on lyso-Gb1 levels, which becomes less robust over time after the maximal change rate [8,9,10,15]. Similarly, we show a more considerable lyso-Gb1 change in those with pre-ERT levels compared to when both lyso-Gb1 levels are on-ERT. Still, lyso-Gb1 levels can also decrease after a prolonged period of therapy. Follow-up on therapy can reflect treatment response and detect treatement failures and compliance issues.

In children on ERT, who achieved a normal platelet count, lyso-Gb1 levels correlated only with hemoglobin levels; a parameter with a slower response to treatment. Similar findings were reported in the splenectomy era; in splenectomized patients, anemia was more prominent than thrombocytopenia, whereas, in non-splenectomized patients, thrombocytopenia is typically more prominent than anemia [3].

In a study which included treated patients, mainly adults, age had a significant inverse correlation with plasma lyso-Gb1 [12]. The authors concluded that this likely reflects the fact that children have a more severe disease at presentation than adults. In our cohort, age was not a predictor of lyso-Gb1 levels in either treated or untreated children. This may be explained by the changing phenotype of children with GD due to pre-natal screening [6,7] or by the fact that only children were included in our study.

Interestingly, age did play a role in the pattern of change in lyso-Gb1 levels over time in mildly effected untreated children. A younger age at first visit was associated with an increase in lyso-Gb1 levels. At an older age, the lyso-Gb1 levels may plateau or even mildly drop. Extended follow-up into adulthood is needed for those untreated children. The fact that lyso-Gb1 could drop without treatment suggests that only a consistent and significant increase in lyso-Gb1 levels should lead to treatment decisions in children presenting with asymptomatic/mildly symptomatic disease.

The additional interesting aspect of our study is the similarity between GD1 and GD3 concerning lyso-Gb1 levels. This might reflect the fact that the GD3 patients in our center belong to the so-called Type 3b (patients with severe visceral manifestation and relatively minimal neurological abnormalities [16]). Others are Type 3c (the so-called "cardiac variant" wherein there is minimal visceral involvement and minimal neurological features but massive calcifications of the aortic and mitral valves [17,18]).

## 4. Material and Methods

### 4.1. Patients Samples

Eighty-one consecutive children (<18 years) with GD who visited the Gaucher Unit at Shaare Zedek Medical Center from July 2014 to December 2018 were included in this study. Lyso-Gb1 measurement had been included in the routine clinical and laboratory assessment during all follow-up (annual/semiannual) visits.

Disease severity for this study was based on genotypes for GD1, i.e., mild GD1 were all children the with N370S/N370S or N370S/R496H genotypes and severe GD1 were N370S compound heterozygotes [19]. Type 3 (neuronopathic disease) was based on clinical phenotype, all with genotypes known to be associated with GD3. Demographic, baseline, and follow-up clinical and laboratory data were extracted from the medical records. The multiple of normal (MN) was calculated for spleen volume and liver volume based on three dimensions by ultrasonography [20,21]. Dried blood spots (DBS) were collected on filter cards (CentoCard®, Centogene, Rostock, Germany) and lyso-Gb1 analysis was carried out in Centogene, Rostock, Germany. Lyso-Gb1 levels were measured using liquid chromatography-mass spectrometry of DBS samples (Centogene AG, Rostock, Germany) as previously described [8]. The study was approved by the Institution Ethics Committee (0291-18-SZMC, 11/2018).

### 4.2. Statistical Methods

For data analysis, we defined in our study three cohorts. The first was an analysis of data at last visit for all patients (*n* = 81), the second was an analysis of data at first visit for untreated patients and patients with pre-treatment data (*n* = 38), and the last analysis was of patients data with more than one visit (*n* = 68). The delta of every measurement between the last and the first visit was calculated. Log transformation of lyso-Gb1 was done to achieve normal distribution. Results are presented as median and range. Correlations between the lyso-Gb1 levels and continuous measurements were tested by non-parametric Spearman's correlation and Pearson's correlation, for non-normal and normally distributed data, respectively. Mann–Whitney non-parametric test was used to compare non-normally distributed data in independent samples. Multivariate analysis was used to clarify the impact of ERT vs. non-ERT and impact of genotypes (mild vs. severe GD1) on the calculated correlation between the log of lyso-Gb1 and the other measurements. IBM SPSS version 25 was used for analysis. Results were considered to be statistically significant when two-tailed *p*-values were ≤0.01.

## 5. Conclusions

The correlations found between lyso-Gb1 levels and disease severity as well as the changes in levels with and without therapy support the importance of using this biomarker in monitoring children with GD, both untreated and treated, for the need to start treatment and to follow the response to therapy, respectively. At this point, we cannot recommend a specific cut-off for lyso-Gb1 levels as a sole indication for beginning ERT, in part because by the time lyso-Gb1 levels became available many children were already on ERT. We believe that within the coming years, with a more significant number of naïve patients, we will be able to define a specific value. In the interim, we strongly recommend including lyso-GB1 in the routine follow-up of all children with GD and a progressive increment in lyso-Gb1 should lead to consideration of ERT in untreated children or to a dose increase in treated children.

## Figures and Tables

**Figure 1 ijms-20-03033-f001:**
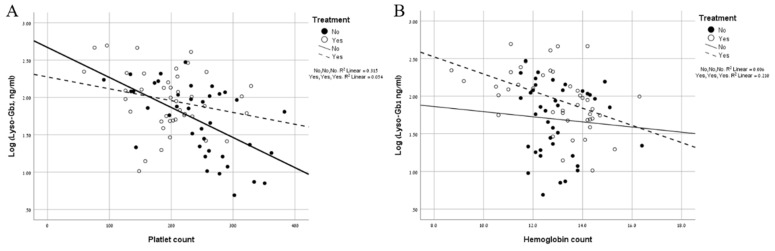
(**A**) Correlation analysis between lyso-Gb1 levels and the platelet count in treated (hollow dots, dotted line) and untreated children (full dots, continuous line). (**B**) Correlation analysis between lyso-Gb1 levels and the hemoglobin level in treated (hollow dots, dotted line) and untreated children (full dots, continuous line).

**Figure 2 ijms-20-03033-f002:**
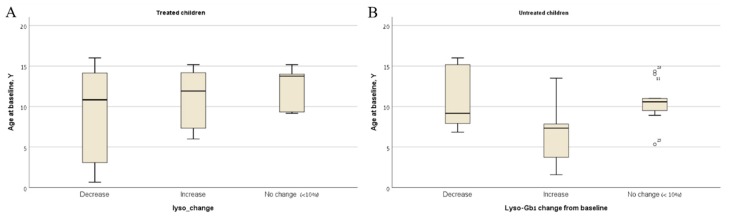
Age at first visit of the children with decreased, increased, and unchanged lyso-Gb1 levels (10%) between first and last visit for (**A**) treated children (**B**) untreated children.

**Table 1 ijms-20-03033-t001:** Epidemiologic and clinical phenotype of the study cohort at the last visit.

	Total	Mild Type 1	Severe Type 1	Type 3
N	81	35	34	12
Age, years*	11 (1–18)	11 (4–18)	12 (2–16)	9.5 (1–18)
Male, %	38 (47%)	16 (38%)	18 (52%)	4 (33%)
ERT*	42 (51%)	2 (5.7%)	30 (88%)	10 (83%)
Platelet count, ×10^3^/mL	214 (59–383)	251 (134–383)	209 (76–334)	190 (59–322)
Hemoglobin, mg/dl	12.9 (8.7–16.4)	12.8(11.5–16.4)	13.15 (9.2–15.3)	12.4 (8.7–16.3)
Spleen (MN)*	1.3 (0–16.7)	1.3 (0.5–4.7)	1.3 (0.6–10.3)	3.4 (1.2–16.7)
Liver (MN)*	1.4 (0.3–3.6)	1.4 (1–2.3)	1.3 (0.3–2.8)	2.3 (1.3–3.5)
Lyso-Gb1 level, ng/mL*	76.3 (4.9–495)	64 (4.9–208)	98 (7.3–495)	100.4 (21.4–210)
Weight, Kg*	37.8 (10.1–76)	34.1 (14.7–70.9)	49 (15.4–76)	29.8 (10.1–52.2)

*, median (range); ERT, Enzyme replacement therapy; MN, multiple of normal; Kg, kilogram.

**Table 2 ijms-20-03033-t002:** Clinical characteristics and laboratory levels at first visit of ten children with pre-treatment measurements and all untreated children.

	Pre-Treatment	Un-Treated
N	10	28
Age, years	5.5 (2–14)	8.5 (1–16)
Male, %	5 (55%)	11 (37%)
Platelet count, ×10^3^/mL	82.5 (68–228)	236.5 (117–339)
Hemoglobin, mg/dl	11.1 (6.7–12.4)	12.7 (11.1–15.7)
Spleen (MN)*	3.9 (1.1–22.9)	1.35 (0.5–5.2)
Liver (MN)*	2.4 (1.2–4.5)	1.7 (1–3)
Lyso-Gb1 level, ng/mL*	262.5 (101–1270)	61.45 (6.1–157)

*, median (range); MN, multiple of normal.

**Table 3 ijms-20-03033-t003:** Pattern of change in lyso-Gb1 (ng/mL) levels from baseline to last measurement in untreated children, treated children with pretreatment measurements and in treated children with both measurements on therapy.

	Untreated	Treated, Pretreatment Baseline	
		Yes	No
N	28	10	30
Male	11	5	16
Age, years*	12 (4–18)	8.5 (3–18)	16 (3–19)
Months of follow-up*	31.85 ( 6.7–45)	27.6 (6.7–44)	28.75 (9.3–49.9)
Number of visits	3 (2–6)	4 (3-6)	4 (2–9)
Unchanged** (n)	9 (32%)	1 (10%)	5 (16%)
Increased (n)	13 (46%)	0 (0%)	8 (26%)
Increase change*	12 (1.29–128)		67.4 (5.7–368)
Decreased (n)	6 (21%)	9 (90%)	17 (56%)
Decrease change*	11.2 (4–50.4)	143.6 (13–1207.7)	32.7 (4.2–172)

*, median (range); **, <10% change from baseline; n, number.

**Table 4 ijms-20-03033-t004:** Clinical and laboratory characteristics of eight children whose lyso-Gb1 increased on enzyme replacement therapy.

Age (Y)*	Gender	Genotype	Mo. on Tx*	Dosa u/kg/mo*	Follow Up (mo)	Baseline Lyso- Gb1	Change from Baseline					Possible Explanation
							Lyso-Gb1	PLT	Hb	Spleen MN	Liver MN	
9	male	Severe GD1	72		36.7	140	79↑	9↓	0.5≈	0.5≈	0.4≈	
9	male	Severe GD1	81.2	52	23.6	168	18↑	31↓	0.4≈			Weight gain**
18	male	Severe GD1	92.3	42	18.6	95	368↑	24↓	1.1↓	1.8↑	0.3≈	Compliance
18	female	Severe GD1	130.9	35	36.9	281	180↑	18↓	0.5≈	0.6↓	0.1≈	
10	male	Severe GD1	90.3	42	40.0	164	48↑	56↓	1.1↑	1.4↓	0.2≈	
14	male	Severe GD1	113.7	114	38.3	45	6↑	4≈	1.3↓	6.4↑	1.2↓	Weight gain**
16	male	Severe GD1	137.0	50	30.2	124	77↑	20↑	2.5↓	0.6↓	0.3≈	Weight gain**
17	male	GD3	124.8	60	18.6	32	13↑	20↓	1.3↑	0.5↑	0.8↑	

Y, year; mo, months; PLT, platelets; Hb, hemoglobin; *, at last visit; **, more than 15% increment in the weight percentage.

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
