# Peer review of "Glucosylsphingosine (lyso-Gb1) as a Biomarker for Monitoring Treated and Untreated Children with Gaucher Disease"

_ijms, 2019, doi:10.3390/ijms20123033_

Round 1

Reviewer 1 Report

This paper is a communication focusing on the use of glucosylsphingosine (lyo-Gb1), a metabolite generated in the catabolism of glucocerebroside (glucosylceramide), as a biomarker to monitor the clinical course/severity in children affected with type 1 non-neuronopathic and type 3, subacute neuronopathic Gaucher disease (GD). Gaucher disease is a sphingolipid storage disease resulting from an inherited deficiency of the lysosomal enzyme glucocerebrosidase. In normal individuals, the level of lyso-Gb1 is low and insignificant. In GD patients, the undegraded glucocerebroside substrate accumulates excessively because of gluocerebrosidase deficiency, and is eventually deacylated to form the cytotoxic lyso-Gb1. Lyso-Gb1 was previously reported as a useful biomarker for monitoring adult GD patients and this brief communication extends the study by analysing the level of lyso-GB1 and reviewing the clinical charts of 81 GD children with mild to severe GD. In this reviewer’s opinion, this work is original and acceptable for publication, as the data presented are relevant and significant, and will contribute to the wealth of information on the application of biomarkers to monitor GD clinical course and consideration of treatment options, e.g. enzyme replacement therapy in patients. The followings are suggested to strengthen its presentations:

Line 50 – The correct spelling is `Glucosylsphingosine’ and not Glucosylsphingosin’

Section 3 Discussion, 2nd paragraph (line 111) - The authors stated that the severity of the GD patients was being determined by genotyping. However, the genotyping data were not shown in the main text nor in the supplementary section. While it is correct that the GD homozygous N370S mutation is usually associated with mild GD phenotype, nevertheless, there is clinical variability, sometimes even among affected sibs with the same genotype due to other factors including modifying genes (Davidson et al Human Mutation 39 (12), 1739-1751, 2018).  

Line 116 – The authors stated that `In treated children, the lyso-Gb1 was associated with compliance ….’.

Did they mean that the increase was due to lack of compliance in those patients?   

Author Response

Reviewer 1:

This paper is a communication focusing on the use of glucosylsphingosine (lyo-Gb1), a metabolite generated in the catabolism of glucocerebroside (glucosylceramide), as a biomarker to monitor the clinical course/severity in children affected with type 1 non-neuronopathic and type 3, subacute neuronopathic Gaucher disease (GD). Gaucher disease is a sphingolipid storage disease resulting from an inherited deficiency of the lysosomal enzyme glucocerebrosidase. In normal individuals, the level of lyso-Gb1 is low and insignificant. In GD patients, the undegraded glucocerebroside substrate accumulates excessively because of gluocerebrosidase deficiency, and is eventually deacylated to form the cytotoxic lyso-Gb1. Lyso-Gb1 was previously reported as a useful biomarker for monitoring adult GD patients and this brief communication extends the study by analysing the level of lyso-GB1 and reviewing the clinical charts of 81 GD children with mild to severe GD. In this reviewer’s opinion, this work is original and acceptable for publication, as the data presented are relevant and significant, and will contribute to the wealth of information on the application of biomarkers to monitor GD clinical course and consideration of treatment options, e.g. enzyme replacement therapy in patients. The followings are suggested to strengthen its presentations:

Line 50 – The correct spelling is `Glucosylsphingosine’ and not Glucosylsphingosin’.

Corrected.  

Section 3 Discussion, 2nd paragraph (line 111) - The authors stated that the severity of the GD patients was being determined by genotyping. However, the genotyping data were not shown in the main text nor in the supplementary section.

Information regarding the genotyping is supplemented in the main text under methods section (section 4.1): 

"Disease severity for this study was based on genotypes for GD1, i.e., mild GD1 were all children with N370S/N370S or N370S/R496H genotypes and severe GD1 were N370S compound heterozygotes. Type 3 (neuronopathic disease) was based on clinical phenotype, all with genotypes known to be associated with GD3".  

 While it is correct that the GD homozygous N370S mutation is usually associated with mild GD phenotype, nevertheless, there is clinical variability, sometimes even among affected sibs with the same genotype due to other factors including modifying genes (Davidson et al. Human Mutation 39 (12), 1739-1751, 2018).  

We agree that there is clinical variability within the same genotype, as also can be seen by the range of Gaucher related parameters in children from the same genotype group. We have added the ref. suggested - row 129-123. We also added in the methods (raw 176) the ref. showing the rational for defining severity by genotype (Zimran, et al. Lancet 2:349-52, 1989).

Line 116 – The authors stated that `In treated children, the lyso-Gb1 was associated with compliance ….’.

Did they mean that the increase was due to lack of compliance in those patients? 

As shown in table 4 – out of 8 patients who had increased levels of lyso-Gb1 during treatment, one patient had obvious compliance issues, and three patients gained weight without a dose adjustment. Therefore we stated that the lyso-Gb1 increase was associated with compliance and lower ERT dose/kg.

 Table 4 was cut- a correct one is enclosed.

Reviewer 2 Report

The manuscript by Hurvitz et al analyzes Lyso-Gb1 levels in treated and untreated children with Gaucher Disease showing:  1) higher levels in children with severe GD1 compared to those with mild GD1; 2) an association between lyso-GB1 levels and clinical status (platelet count) in untreated children and 3) children who started treatment based on clinical criteria had higher levels of lyso-GB1 than children who were not treated.  Based on their findings, the authors suggest the use of lyso-GB1 as a biomarker for monitoring children with GD and suggest that a progressive increment in lyso-GB1 should lead to consideration of ERT initiation.

The manuscript is written by a top leader group in the area and provides data from a large cohort of children with Gaucher Disease and will be interesting for the readers. However, some issues should be addressed:

1.- In the manuscript it is stated that “Multivariable analysis is used” between lyso-GB1 levels and other variables (platelet and Hemoglobin counts) in treated and untreated patients. Multivariable analysis refers to regression analysis, while in this manuscript bivariate analysis (Pearson or Spearman correlation) is used in two different groups of patients (treated and untreated).

2.- In line 60 there is a p value of “1.81”. Please explain or correct.

3.- In lines 67 and 68, along with the p values, correlation coeficients should be included.

4.- Both Figure 1 and Figure 2 are very small and can´t be seen properly.

5.- Please include p values in the Figures.

6.-  In  Figure 1 it says “No, no, no” and “yes, yes, yes”. Please explain what this refers to.

7.- Figure 1 legend should be rewritten. Some sentences lack a verb.

8.- Give the results in the “Results section” and not in the Figure legends (both Figure 1 and 2).

9.- The use of English should be revised to correct some minor mistakes.

10.- Please read carefully and correct typos like in line 85 where the “and” between treated and untreated children is missing.

11.- In line 178 it says that “results were considered to be statistically significant when two-tailed p values were <= 0.01. Why 0.01 and not 0.05 as usually accepted?

Author Response

Reviewer 2

The manuscript by Hurvitz et al. analyzes Lyso-Gb1 levels in treated and untreated children with Gaucher Disease showing:  1) higher levels in children with severe GD1 compared to those with mild GD1; 2) an association between lyso-GB1 levels and clinical status (platelet count) in untreated children and 3) children who started treatment based on clinical criteria had higher levels of lyso-GB1 than children who were not treated.  Based on their findings, the authors suggest the use of lyso-GB1 as a biomarker for monitoring children with GD and suggest that a progressive increment in lyso-GB1 should lead to consideration of ERT initiation.

The manuscript is written by a top leader group in the area and provides data from a large cohort of children with Gaucher Disease and will be interesting for the readers. However, some issues should be addressed:

1.- In the manuscript, it is stated that “Multivariable analysis is used” between lyso-GB1 levels and other variables (platelet and Hemoglobin counts) in treated and untreated patients. Multivariable analysis refers to regression analysis, while in this manuscript, bivariate analysis (Pearson or Spearman correlation) is used in two different groups of patients (treated and untreated).

The analysis done in figure 1 was multivariable - a linear regression for lyso-Gb1, including both the platelet/hemoglobin values and treatment status in the model. Similarly was done for adjusting for genotype.

2.- In line 60 there is a p-value of “1.81”. Please explain or correct.

Thank you for the comment. It was corrected. 

3.- In lines 67 and 68, along with the p values, correlation coefficients should be included.

P values and correlation coefficients were added.

4.- Both Figure 1 and Figure 2 are very small and cannot be seen properly.

Corrected. 

5.- Please include p values in the Figures.

P values were added.

6.-  In  Figure 1 it says, “No, no, no” and “yes, yes, yes.” Please explain what this refers to.

The "no no no" and "yes yes yes" were written automatically by the SPSS program and are not necessary for the figure understanding. We enclose a corrected figure.   

7.- Figure 1 legend should be rewritten. Some sentences lack a verb.

Corrected. 

8.- Give the results in the “Results section” and not in the Figure legends (both Figure 1 and 2).

Corrected. 

9.- The use of English should be revised to correct some minor mistakes.

Done

10.- Please read carefully and correct typos like in line 85 where the “and” between treated and untreated children is missing.

Done

11.- In line 178 it says that “results were considered to be statistically significant when two-tailed p values were <= 0.01. Why 0.01 and not 0.05 as usually accepted?

A p-value of 0.01 was chosen to reflect the number of analysis done in a relatively small cohort of patients.  
